# Twenty-Year Survival of Patients Operated on for Non-Small-Cell Lung Cancer: The Impact of Tumor Stage and Patient-Related Parameters

**DOI:** 10.3390/cancers14040874

**Published:** 2022-02-10

**Authors:** Olivier Schussler, Antonio Bobbio, Hervé Dermine, Audrey Lupo, Diane Damotte, Yves Lecarpentier, Marco Alifano

**Affiliations:** 1Thoracic Surgery Department, Cochin Hospital, APHP Centre Paris University, 75014 Paris, France; antonio.bobbio@aphp.fr (A.B.); marco.alifano@aphp.fr (M.A.); 2Department of Anesthesia and Surgical Intensive Care, Cochin Hospital, APHP Centre Paris University, 75014 Paris, France; herve.dermine@aphp.fr; 3Pathology Department, Paris Center University Hospitals, AP-HP, University Paris Descartes, 75006 Paris, France; andrey.lupo@apbp.fr (A.L.); diane.damotte@aphp.fr (D.D.); 4Inserm U1138, Integrative Cancer Immunology, University Paris Descartes, 75006 Paris, France; 5Centre de Recherche Clinique, Grand Hôpital de l’Est Francilien (GHEF), 77120 Coulommiers, France; y.c.lecarpentier@aphp.fr

**Keywords:** lung cancer, surgery, outcome, very long-term survival, prognostic factors

## Abstract

**Simple Summary:**

Surgery is the mainstay treatment of non-small-cell lung cancer, but its impact on survival beyond 15 years has never been reported so far. We studied retrospectively clinical characteristics and short and long-term survival of a single-institution patient population whose baseline data were prospectively collected. All patients underwent major lung resection between June 2001 and December 2002. Vital status was obtained by checking INSEE database and verifying if reported as “non-death” by the hospital administrative database and direct phone interviews with patients of families. A total of 345 patients were analyzed; 15-year and 20-year overall survival rates were 12.2% and 5.7%, respectively. At univariate analysis, predictors of worse survivals were increasing age at surgery, lower BMI, weight loss, higher baseline C-reactive protein, pathological stage, and, among patients with adenocarcinoma, higher grade. Increasing age, cumulative smoking, lower BMI, and pathological stage were independent predictors of long-term survival at Cox multivariate analysis. We conclude that very-long-term survivals can be achieved after surgery of NSCLC, and factors classically predicting 5- and 10-years survival also determines longer outcomes suggesting that both initial tumor aggressiveness and host characteristics act beyond the period usually taken into account in oncology.

**Abstract:**

Surgery is the mainstay treatment of non-small-cell lung cancer (NSCLC), but its impact on very-long-term survival (beyond 15 years) has never been evaluated. Methods: All patients operated on for major lung resection (Jun. 2001–Dec. 2002) for NSCL in the Thoracic Surgery Department at Paris-Hôtel-Dieu-University-Hospital were included. Patients‘ characteristics were prospectively collected. Vital status was obtained by checking INSEE database and verifying if reported as “non-death” by the hospital administrative database and direct phone interviews with patients of families. Results: 345 patients were included. The 15- and 20-year survival rates were 12.2% and 5.7%, respectively. At univariate analysis, predictors of worse survivals were: increasing age at surgery (*p* = 0.0042), lower BMI (*p* = 0.009), weight loss (*p* = 0.0034), higher CRP (*p* = 0.049), pathological stage (*p* = 0.00000042), and, among patients with adenocarcinoma, higher grade (*p* = 0.028). Increasing age (*p* = 0.004), cumulative smoking (*p* = 0.045), lower BMI (0.046) and pathological stage (*p* = 0.0026), were independent predictors of long-term survival at Cox multivariate analysis. In another model, increasing age (*p* = 0.013), lower BMI (*p* = 0.02), chronic bronchitis (*p* = 0.03), lower FEV1% (*p* = 0.00019), higher GOLD class of COPD (*p* = 0.0079), and pathological stage (*p* = 0.000024), were identified as independent risk factors. Conclusions: Very-long-term survivals could be achieved after surgery of NSCLC, and factors classically predicting 5- and 10-years survival also determined longer outcomes suggesting that both initial tumor aggressiveness and host’s characteristics act beyond the period usually taken into account in oncology.

## 1. Introduction

Lung cancer is the leading cause of cancer death worldwide [1,2,3,4]. With respect to treatment, surgical resection is generally proposed at the less advanced stages of the malignancy and has been shown to contribute mainly to improved survival rates up to 5 years (after treatment), especially if a multimodal approach is used [5]. Surprisingly, unlike in other malignancies (hematologic disorders, for example), survival after lung cancer treatment (i.e., operated or not) has only been investigated for a relatively short period of time, so that information is only available for 5-year survival and, in a limited number of studies, 10-year survival [6,7,8,9].

With regard to long-term outcomes after surgical resection, studies on 10-year survival rates have only been made by single institutions [6,7], and only two institutional studies involving small samples have investigated longer-term survivals, of 13 and 14 years, respectively [8,9]. In these studies, factors associated with longer survival were younger age, female sex, lower tumor stage, and lobar resection (versus pneumonectomy) [9]. Another study, which included smoking habits, identified the following independent risk factors for 7-year survival: lower pathological stage, smoking cessation before surgery, lower age, adenocarcinoma histology, and lobar resection; sex of patient was not a factor [10].

The impact of other patient-related factors is currently gaining interest: thinner corpulence [11,12] or sarcopenia [11,13] have been shown to negatively affect 11-year survival after lobectomy [7]. On the other hand, body overweight (BMI ≥ 25 Kg/m^2^) and obesity (BMI ≥ 30 Kg/m^2^) may play a protective role at 5 and 7 years [12,14,15,16]. Although obesity is a risk factor for carcinogenesis in at least 20 cancer types, epidemiological studies have consistently demonstrated a lower risk of developing lung cancer and of dying after treatment in overweight/obese patients, referred to as the “lung cancer obesity paradox” [17,18]. The impact of obesity or being overweight on very long-term survival rates in lung cancers warrants specific investigation.

Respiratory functional parameters have also been assessed as determinants of 5-year and 10-year survival: in particular, in studies including patients with severe COPD and/or emphysema, altered pulmonary function, measured as lower FEV1, has been shown to be an independent predictive factor at 5 [19], 7 [20], and 10 years [10,20]. When focusing on COPD patients only, the severity of disease has been shown to be associated with higher mortality at the same time points [6,21]. Restrictive patterns, as in interstitial lung diseases, have also been shown to be associated with an increased risk of lung cancer [22,23] and a poorer 5-year prognosis after lung resection [24].

Apart from institutional studies, some registries have also reported on the long-term survival rates of resected lung cancer patients: two European registries, namely the Norway and the French Epithor databases, reported on 5-year and 11-year survivals, respectively [12,25], while North American registries assessed figures at 10 years [26] and, for patients older than 65 years, at 14 years [25,27,28]. An analysis based on the database of the Society of Thoracic Surgeons (STS) revealed the following factors to be associated with 7-year survival: increased age, male sex, smoking history, BMI (BMI < 18.5, 18.5 ≤ BMI < 25, 25 ≤ BMI < 30, 30 ≤ BMI < 35, BMI ≤ 35), and decreased preoperative forced expiratory volume in 1 s. (percentage of normal value FEV1% (FEV1% < 40%, 40% ≤ FEV1% < 60%, 60% ≤ FEV1% < 80, 80% ≤ FEV1%) [20]. Overall, these registries have the great advantage of including a huge number of patients but lack details in the reported clinical features: for example, important risk factors such as cumulative tobacco consumption have not been investigated, and smoking habits were only assessed in a qualitative manner.

Thus, the objective of this study was to assess, in the context of a single institution, the long-term survival rates (up to 20 years) of patients after surgery for NSCLC, a subject that, to the best of our knowledge, has not been reported on so far. An additional objective of the study is to identify risk factors, focusing not only on tumor-related characteristics or basic demographic features but also on a more detailed assessment of patient comorbid illnesses. The availability of a prospectively collected database in our own institution allowed us to perform this work.

## 2. Materials and Methods

### 2.1. Patient Inclusions and General Management

This study is based on retrospective analysis of a prospectively collected database. The database provided the demographic, morphometric, and clinical characteristics of all consecutive patients that underwent major lung resection for a lobectomy or pneumonectomy between 15 June 2001, and January 2002 for NSCLC in a single high-volume university hospital (Cochin-Hôtel Dieu hospital Paris, France) were included. Patients with carcinoid tumors and small-cell lung cancer were excluded. Patients surgically treated with sublobar resections (wedges or segmentectomies) were not included to avoid confounding factors. Thus, 393 patients were enrolled for the present study, originating mainly from France. The same cohort has been used for previous studies on post-operative pneumonia and to assess 5- and 7-year survival rates after resection for non-small-cell lung cancer [14,29,30].

For all patients, the preoperative workup included a chest X-ray, a fiberoptic bronchoscopy, a full-body contrast-enhanced CT scan. An isotopic positron emission tomography was not available and not recommended for all patients at the time of patient inclusion. Invasive mediastinal staging was carried out by cervical mediastinoscopy/anterior mediastinotomy or video-assisted thoracoscopy if enlarged nodes (i.e., short axis > 1 cm) were seen on the CT scan, and neoadjuvant platinum-based chemotherapy (2 to 4 cures) was prescribed if N2 disease could be pathologically proven. In this subset of patients, an assessment of responses was performed by a CT scan, and patients were offered surgery in the event of clinical response or stable disease. For stage IV diseases, only oligometastatic cases were considered eligible for surgery, provided that a lobectomy could be anticipated and no mediastinal nodal involvement was present. At this time, all the patients underwent major pulmonary resection by a classical open posterolateral thoracotomy. A full nodal dissection was carried out in all cases. Adjuvant radiotherapy or chemotherapy was proposed on an individual basis following evidence-based discussions in a multidisciplinary team, under the supervision of the referring pneumologist or oncologist.

For all cases, a pathological review of the samples was performed by two expert pathologists. All adenocarcinoma cases were reclassified independently according to the IASLC/ATS/ERS classification [31] based on the predominant architectural pattern. Thus, adenocarcinomas were classified into three grades, as previously described [32]. Upon pathological reviewing, tumor stages were reattributed according to the 7th edition of the TNM classification [33,34].

IRB approval was obtained (i.e., Comité de Protection des Personnes (CPP); n°208-133 and 12 June 2012). The IRB dispensed us with obtaining patient informed consent because of the retrospective, non-interventional character of the study and the high number of deceased patients when the study was performed. Patient records were anonymized and de-identified prior to analysis. The study was conducted according to the recommendations outlined in the Declaration of Helsinki and French laws on biomedical research.

### 2.2. Collected Data

Patient characteristics, treatment procedures, and short-term outcomes have been prospectively collected using a standardized case report form (CRF) as previously mentioned [29]. For the study population, we collected demographic data, patient history and clinical characteristics (comorbidities, tobacco or alcohol consumption, nutritional status, including BMI, weight loss, preoperative prealbumin, and preoperative CRP levels, respiratory functional assessment, including preoperative FEV1% and GOLD stage), treatment modalities (including induction treatment and extent of surgery), and pathologic features (histologic type, and, for adenocarcinoma, grading and pathologic stage). We also collected in-hospital and 30-day mortality.

With respect to very long-term survival, the same cohort was used as in previous studies with a 5- to 7-year follow-up assessment [14,32,35]. We updated data on later vital status by direct interrogation (in April 2021) of the French INSEE (Institut National de la Statistique et des Etudes Economiques) open-source national database (i.e., open-link Internet), which is already used for the assessment of lung cancer survival after surgical treatment by the French Epithor team [12]. The INSEE database is not exhaustive for foreign patients (even those having spent a long period in France), so among those patients born outside France, we considered as dead those stated as dead in the INSEE database, but we excluded the remaining cases from the study (*n* = 25).

For patients born in France, similarly, we considered as dead those stated as dead in the INSEE database (with the relevant death date), but checked the vital status of the remaining patients by: (1) interrogation of electronic clinical files of our center; (2) interrogation of the clinical data system shared by all the university hospitals in Paris and the neighboring area (APHP); (3) direct phone interview with patient or relatives. This allowed us to exclude 4 patients (telephone or cellphone attributed but no response to iterative calls). Finally, we also excluded 19 French patients who had been totally lost between the last follow-up interview performed after 7 years and the new 20-year follow-up assessment.

### 2.3. Statistical Analysis

Data processing and analysis were performed with the statistical Software SEM (Silex Development, Mirefleurs, France). Results are expressed as percentages and mean +/− SD for normally distributed quantitative variables and as median interquartile ranges for non-normally distributed quantitative variables.

Survival analyses were performed using the Kaplan–Meier method, and curves were compared with the log-rank method. Most factors used in the univariate analysis were entered in a stepwise multivariate Cox proportional hazards model, with censoring at 20 years, to assess their independent character. Several models have been developed to assess several clinical variables in addition to biological variables.

## 3. Results

### 3.1. Clinical Features

The study population consisted of 345 patients. Their demographic, clinical, and morphologic parameters are detailed in Table 1. As usual, at the time of initial inclusion, patients were more frequently men (84%), with a current or past history of tobacco smoking (93%), with cumulative smoking mean of 50 packs/year. The most frequently reported comorbid illnesses were chronic bronchitis, alcohol abuse, and diabetes mellitus in 64%, 26%, and 13% of cases, respectively. A total of 44% of cases fulfilled the definition of COPD. With respect to preoperative nutritional parameters, 10% had a body mass index (BMI) of <18.5 Kg/m^2^, and 38% had a BMI ≥ 25 Kg/m^2^. A total of 65% of patients had a stable body weight or a decrease of <5%, 16% had a weight loss between 5% and 10%, and 18% had a weight loss of more than 10%. A total of 43% of patients had a very low preoperative CRP of ≤3 mg/mL. With respect to preoperative treatments, 23% received chemotherapy and 4% radiotherapy. Lobectomy was more frequently carried out. The histological types were adenocarcinoma in 47% of cases (i.e., with 40% of them being grade 3), squamous cell carcinoma in 39.8%, and large-cell carcinoma in 7% of patients. The most common disease stage was I (41%), followed by III (32%) and II (21%). Only 4% of patients had oligometastatic stage IV.

### 3.2. Long-Term Outcomes

The median survival of the whole study population (*n* = 345) was 41 months. The 5-, 10-, 15-, and 20-year survival rates were 41.9% (36.7–47.1), 23.8% (19.6–28.6), 12.2% (9.1–16.0), and 5.7% (3.4–9.3), respectively (Table 2). Figure 1 reports the Kaplan–Meier survival curves for the whole population. For comparison after 10 years, for the time-frames 10–15 years and 15–20 years, the mortality was very similar to that observed for a normal control population of patients of the same age but without tumor (see dotted green line in Figure 1 as reported by Batevik R. et al. *Lung Cancer 2005* [9]).

Table 2 shows the results of univariate survival analyses of 16 relevant clinical parameters. In our study at 20 years, increased age was strongly associated with poorer long-term survival (*p* = 0.0042) (Figure 2(A2)), while only a weak link with the sex of the patient (*p* = 0.059) was observed in this data set (Figure 2(A1)). With regard to tobacco consumption, neither smoking status at surgery (*p* = 0.24) nor smoking cessation before surgery (*p* = 0.39) impacted the outcome. At the same time, chronic bronchitis (*p* = 0.027) (Figure 2(E1)) and cumulative tobacco consumption (*p* = 0.021) (Appendix A) were associated with a poorer survival. By univariate analysis, the following were also not found to be significantly associated with very long-term survival: preoperative FEV1 in percentage of predicted value (FEV1%) (*p* = 0.82) (Appendix A), the presence of COPD (*p* = 0.26) (Appendix A) and its severity in terms of GOLD category (*p* = 0.94) (Appendix A). Regarding additional comorbid illnesses, a weak association (not statistically significant) with outcome was observed for alcohol abuse (*p* = 0.062) (Appendix A) but not diabetes mellitus (*p* = 0.31) (Appendix A).

With respect to inflammation and nutritional parameters, body mass index (BMI) was significantly associated with outcome, with patients with the lowest BMI < 18.5 having the poorest outcomes, while patients with a higher BMI (25 ≤ BMI) being protected (*p* = 0.0090) (Figure 2(F1)). Similarly, the importance of weight loss in the last 6 months was also associated with a poorer outcome (*p* = 0.0034) (Figure 2(F2)). On the other hand, preoperative prealbumin ≥ 275 mg/mL (shown to influence survival up to 7 years [32] in an independent manner) was also associated with a slight trend (non-significant) toward better survival (*p* = 0.065) (Figure 2(G2)). Interestingly, in this study, Preoperative C-reactive protein (CRP), which was shown to influence survival at 7 years in an independent manner [32], remained a positive predictive factor at 20 years by univariate analysis (*p* = 0.049) (Figure 2(G1)).

At 20 years, the most significant factors were: the pathological stage of the tumor (whether expressed in four classes, (i.e., I, II, III, and IV (*p* = 0.00000042) (Figure 2(B1))), or in seven classes, (i.e., IA, IB, IIA, IIB, IIIA, IIIB, or IV (*p* < 0.0000001) (Appendix A)) and interestingly also the “classical factors” historically shown to predict 5- and 10-year survival after surgery. Although histologic type (adenocarcinoma, squamous cell carcinoma, or large-cell carcinoma) was not significantly associated with poorer survival periods in our study population (*p* = 0.33) (Figure 2(C1)), among patients with adenocarcinomas, those with high-grade adenocarcinomas had poorer outcomes (*p* = 0.028) (Figure 2(C2)). Finally, less extended resections (lobectomy or bilobectomy) were associated with significantly better long-term survival than those relating to pneumonectomy (*p* = 0.0000067) (Figure 2D).

#### 3.2.1. Independent Risk Factors (Clinical and Biological) Affecting Long-Term Survival following Multivariate Cox Analyses

##### Independent Clinical Factors

Two Cox models were built. Model 1 included most clinical variables, such as age (<50, 50–60, 60–70, 70 ≤ years), sex, BMI category (underweight, normal weight, overweight), smoking status (current, former, never), cumulative smoking (<20, 20–50, 50–100, 100 ≤ Pack years), history of chronic bronchitis, COPD, GOLD class (1-2-3, (i.e., no GOLD 4 in our study)), diabetes mellitus, alcohol abuse, weight loss (<5%, 5–10%, 10%≤), type of resection (lobectomy/bilobectomy, pneumonectomy), histologic type (squamous, adenocarcinoma, large-cell carcinoma, others), pathologic stage (IA, IB, IIA, IIB, IIIA, IIIB, IV). Model 2 was also a clinical model and similar to model 1 but was adjusted for preoperative FEV1% (<50, 50–60, 60–70, 70–80, 80 ≤ % predicted). The first model showed that age (*p* = 0.069), cumulative smoking (*p* = 0.019), BMI category (*p* = 0.016) and stage of disease (*p* = 0.0011) independently predicted long-term survival, with the extent of resection also showing a tendency slight trend (*p* = 0.096) (Table 3). The second model showed that VEMS category (*p* = 0.00014), stage of disease (*p* = 0.0014), chronic bronchitis (*p* = 0.019), GOLD stage (*p* = 0.045) and weight loss (*p* = 0.018) independently predicted outcomes (Table 3).

##### Independent Risk Factors Affecting Very Late Mortality: Analysis of Clinical-pathological Parameters and Laboratory Parameters

As preoperative laboratory parameters were available for most patients in the series, we built two other models that included both the above parameters (with and without preoperative FEV1%) and inflammatory parameters in the form of preoperative CRP levels (CRP ≤ 3 mg/mL, 3 mg/mL < CRP). As shown in Appendix A, model 3 corresponds to model 1 adjusted for CRP levels. The independents factors were: age (*p* = 0.006), CRP levels (*p* = 0.041), stage of disease (*p* = 0.045), cumulative smoking (*p* = 0.048), and BMI class (*p* = 0.044) while chronic bronchitis also showed a strong, but not statistically significant, trend (*p* = 0.076). In the other model 4, (corresponding to model 2 but also adjusted for preoperative CRP levels), the independent factors were: FEV1% class (*p* = 0.000018), GOLD category (*p* = 0.0025), chronic bronchitis (*p* = 0.012), age (0.0075), stage of disease (*p* = 0.019), and CRP levels (*p* = 0.037). These factors independently predicted outcome, while cumulative smoking (*p* = 0.053) showed a strong tendency but did not reach statistical significance (Appendix A).

## 4. Discussion

In our study, we assessed late survival (up to 20 years) in a large monocentric cohort of patients treated with major lung resection for NSCLC. To the best of our knowledge, such a study is the first of its kind. We showed that overall survival rates at 15 and 20 years were, respectively, 12.2% (9.1–16.0) and 5.7% (3.4–9.3).

We were able to identify the following factors in predicting very late survival: initial tumor stage, age at surgery, chronic bronchitis, COPD, systemic inflammation, and nutritional status, with undernutrition being unfavorable and overweight/obesity a protective factor. All these clinical factors have also been reported as impacting on early mortality at 5 and 10 years; our study, therefore, suggests that their impact may continue later as well.

In this study, we reported a global survival rate of 41.9% at 5 years and 23.8% at 10 years. With respect to the stage of disease (1 to 4), 10-year survival figures were, respectively, 30%, 14%, 12%, and 14%. All these figures are similar to those reported from large cohorts. For comparison, in the STS database that includes 22,000 patients operated for lung cancer, survival rates at 5, 10, and 14 years for stage I-II were, respectively, around 50%, 25%, and 15% [32]. For stage III–IV, survival figures were around 20% at 5 years, 10% at 10 years, and less than 5% at 14 years [32].

Like our findings for the stage of disease, most risk factors identified in our study as predictors of late survival have also been identified as affecting shorter term (5 or 10-year) survival rates, in particular: age [22,31], chronic bronchitis [35], COPD [6,17] FEV1% [10,17,22], smoking history [22] or cumulative smoking [9,10,17,22,31], nutritional status [22], BMI [12], weight loss [14], and systemic inflammation.

With respect to chronic bronchitis/COPD, we previously reported that chronic bronchitis was an independent predictive factor for 5-year survival [32,35]. We have also shown in an earlier study that COPD and chronic bronchitis reduce the number of mature dendritic cells in the tumor microenvironment of NSCLC [32,35], whereas an increased number of these antigen-presenting cells in the tumor microenvironment has been shown to be associated with better survival [6,21,32]. In this study, the effect of chronic bronchitis was an independent negative prognostic factor and persisted, in the model, even in the presence of cumulative smoking, which is the main driver of chronic bronchitis. In our multivariate models, the effect of bronchitis persisted after introducing preoperative CRP, a marker of systemic inflammation, underlining the negative predictive value of both systemic and local inflammation. Chronic bronchitis, also known as “chronic mucus hypersecretion”, is common in cigarette smokers. Chronic bronchitis is a key component of chronic obstructive pulmonary disease (COPD). It has been clearly shown to be linked to COPD disease progression, hospital admission, and mortality [36,37]. Clinically, the presence of the chronic bronchitis phenotype (i.e., as compared with its absence) is associated with an accelerated loss of lung function and an increased frequency of COPD exacerbation [38,39].

Airway mucin (MUC) concentration is a key marker of chronic bronchitis [38,39]. MUC1 and MUC4 isoforms have been associated with tumors. In normal epithelial cells, MUC1 is expressed on the apical side of the cells in a polarized manner. During carcinogenesis, the amount of MUC1 mRNA increases and, due to altered glycosylation, the protein is repositioned, leading to basolateral and cytoplasmic localization [40,41,42]. In models with bronchial epithelial cells, cigarette smoke has been shown to disrupt the integrity of adherens junctions via mucin glycolyzation and EGFR activation [40,41,42]. MUC1 affects cancer progression in lung adenocarcinoma, and its aberrant expression pattern has been correlated with poor tumor differentiation and impaired prognosis. In contrast to normal apical localization of MUC1, a basolateral expression is present in solid high-grade adenocarcinoma [41,43,44]. This is associated with smoking status. Cigarette smoke in vitro increases MUC1 in adenocarcinoma cells [41,43,44]. It has been reported that MUC1 alters integrin binding to the extracellular matrix and promotes further invasion [43,45,46]. Mucin-4 is also normally expressed in the lungs and, like MUC1, exerts its oncogenic effects by altering behavior on lung cancer cells. The level of MUC4 is also elevated in lung cancer and may also play a role [45,47]. Recently it has been shown that chronic bronchitis and smoking are associated with the presence of MUC1 and MUC4 [47,48], as well as sub-epithelial cell fibrosis and airway wall thickening [49,50].

All smokers have increasing amounts of macrophages and neutrophils in their lungs. The presence of this inflammatory component may favor the development of distal or more proximal tumors [49,51]. In COPD patients, lymphocytes are polarized toward a T helper (Th1) phenotype with the secretion of interferon-gamma while in the presence of tumors [50], there is an immune infiltrating cell shift toward the M2 phenotype [50,52]. The role of inflammation in lung cancer has been reviewed recently [52,53]. The immune system plays a critical role in maintaining tissue homeostasis, cell turnover, and tissue remodeling and in preventing infection and cell transformation [52,53]. The inflammatory component in the development of the neoplasm includes a diverse leukocyte population that promotes tumors through the release of a variety of cytokines, chemokines, cytotoxic mediators, such as oxygen species ROSs, metalloproteinase, interleukins, and interferons. Cancer inflammation affects many aspects of malignancy, including proliferation and survival of malignant cells, angiogenesis, and tumor metastasis [52,53].

In this study, we confirmed that obese and overweight patients were protected while patients with lower weight or weight loss had a poorer prognosis. In patients with lung cancer, unlike most other cancers, obesity is associated with improved survival [23,24].

In this study, we demonstrated the impact of weight loss (5% and 10%) and low weight (BMI < 18.5) on long-term outcomes. Weight loss of over 5% is recognized as a classical cause of death for patients with cancer, including patients with lung cancer [37,39,54]. It has also been shown that patients with a weight loss of more than 10% have a significantly lower survival rate than those with a less pronounced weight loss [54,55]. Several studies have shown the impact of a low muscular mass (“sarcopenia”) as an independent factor for the different patients after stratification on BMI [11]. We reported that a low muscular mass was present in 66% of underweight patients with BMI < 18.5, in 39% of normal-weight patients (BMI between 18.5 and 25), only 21% in overweight patients (BMI between 25 ≤ BMI < 30), and rarely in obese patients (BMI ≥ 30) [14]. Sarcopenia was still an independent factor for survival at 7 years. It has been reported that cachexia is an independent factor rather than BMI in predicting survival in patients with lung cancer [11] (for review [13]). Sarcopenia is associated with a poor prognosis in patients with NSCLC at 5 years [42,55] and 7 years [14]. How nutritional status, especially muscular cachexia, is associated with poorer outcomes is not well understood. The lack of body reserves occurring in cachexia may promote a catabolic state favoring cancer development [42]. Furthermore, it is known that malnutrition is a major cause of immune deficiency, promoting infection and cancer development. We have previously reported in resected NSCLC that tumor-infiltrating immune cells, in particular, cytotoxic CD8+ lymphocytes, are directly correlated with a suitable nutritional status [32,35]. From a metabolic point of view, the lack of body reserves may promote a catabolic state of the host) [42,44,46]. Glucogenesis is sustained by glycerol derived from lipolysis, a process that spares and delays the consumption of amino acids derived from proteolysis. Thus, consumption of fat storage sparing proteins would attenuate the loss of skeletal muscles. However, low-fat reserves force cancer cells to consume more and more amino acids in a vicious cycle resulting in low muscle mass, weight loss, and altering the immune response. This perverted metabolism of the host will produce glucose for the proliferation of cancer cells that rely especially on aerobic glycolysis (i.e., the Warburg effect) [44,46].

### Strengths and Limitations of the Study

This study was conducted in a single institution with a low number of patients compared to nationwide databases. At the same time, survival for a period of more than 15 years has never been reported for NSCLC with or without surgery. The follow-up of national databases is limited to only 14 years for the longest observation time. One another important strength of the present study is that we collected prospectively a wide range of variables covering many patient characteristics. In our study cohort from a single institution (the thoracic department of a university hospital), survival was assessed at several earlier time points: 5 years [32,35], 6 years [35,38], and 7 years [14]. Most of the patients in the cohort were followed by our department and institution. Importantly, direct contact with patients or relatives enabled us to obtain true survival figures at each time point, including the 20-year point. Finally, compared with our study, the national databases have only investigated a few patient clinical characteristics, so important information on tobacco-related variables is generally missing [30,31,32].

## 5. Conclusions

Very long-term survival rates can be achieved after surgery for NSCLC. Many clinical factors that predict 5-/10-year survival, such as pathological stage, chronic bronchitis, chronic obstructive diseases (COPD), and pulmonary function (FEV1% (percentage of predictive value)), as well as nutritional parameters, are also determining 20-year outcomes, suggesting that tumor aggressiveness acts beyond the period usually taken into account in oncology.

## Figures and Tables

**Figure 1 cancers-14-00874-f001:**
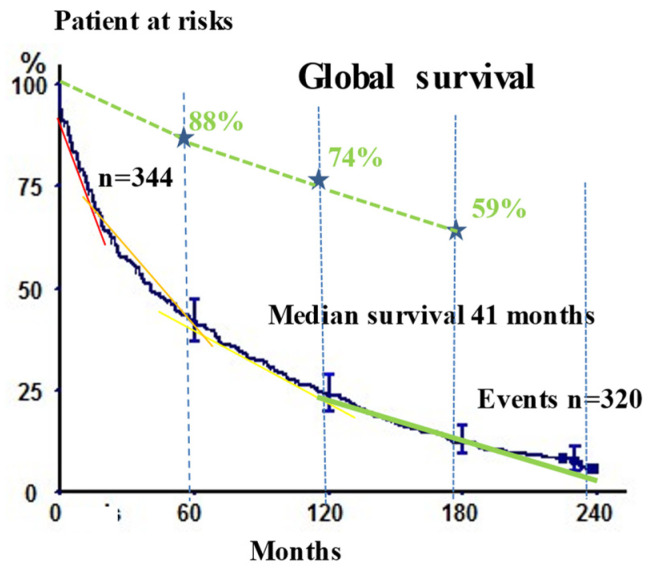
Kaplan–Meier overall survival analyses of the study cohort of 345 patients, with confidence intervals, at 5, 10, 15, and 20 years. Mean age of patients at implantation was 62.1 years. The dotted green lines show the survival rates at 5, 10, and 15 years in the control population of the match for age but without tumor. For this control group of patients, survival rates were: 88% at 5 years, 74% at 10 years, and 59% at 15 years, as reported in an earlier study [9]. Very interestingly, after 10 years, the survival rates in patients having NSCLC (solid green line) were very similar and parallel to people without (dotted green lines). This means that after 10 years and up to 20 years, operated patients had reached the classical mortality rate of non-operated patients.

**Figure 2 cancers-14-00874-f002:**
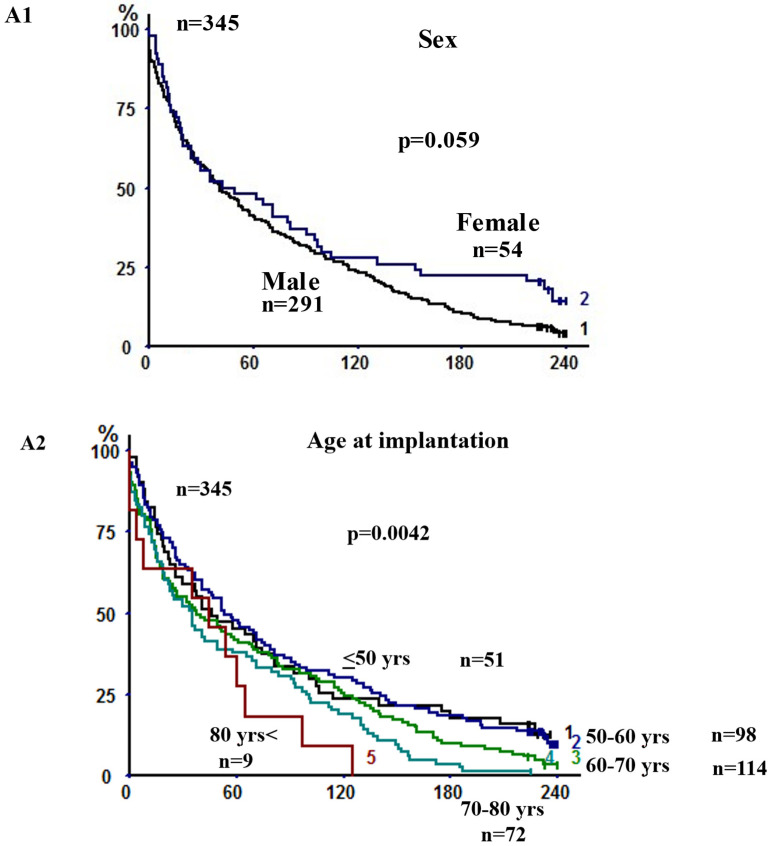
Kaplan–Meier survival analyses and log-rank comparisons for the patient of the study for different variables: sex (panel **A1**), age at implantation (panel **A2**), pathological stages (4 classes) (panel **B1**), tumor characteristics (type of tumor (panel **C1**) and grade of adenocarcinoma (panel **C2**)), type of resection (panel **D**), chronic bronchitis (panel **E1**), morphomics and nutritional parameters (BMI (panel **F1**) and preoperative weight loss (panel **F2**)), biological preoperative parameters (CRP (panel **G1**) and preoperative prealbumin (panel **G2**)). See also Appendix A for: pathological stages (8 classes) (panel **B2**), COPD (panel **E2**), GOLD COPD (panel **E3**), preoperative FEV1% (panel **E4**), cumulative smoking (panel **E5**), alcohol (panel **E6**), diabetes mellitus (panel **E7**).

**Table 1 cancers-14-00874-t001:** Clinical, surgical, pathological, and biological characteristics in the cohort of 345 patients.

Variable	*n (%) or Mean ± SD*
Total	*n* = 345
Sex	
Female, n_0_ (%)	54 (15.7)
Men, n_0_ (%)	291 (84.3)
Age, mean (SD) yrs	62.1 (10.5)
Smoking	
Past or current smoker, n_0_ (%)	319 (93.3)
Smoking cessation before surgery, n_0_ (%)	182 (57.4)
Cumulative smoking: packs/year index, mean (SD)	44.9 (23.6)
<20 pks/yr., n_0_ (%)	21 (7.1)
20–50 pks/yr., n_0_ (%)	155 (86.4)
50–100 pks/yr., n_0_ (%)	109 (36.9)
100 ≤ pks/yr., n_0_ (%)	10 (3.4)
Comorbid illnesses and respiratory status	
Alcohol abuse, n_0_ (%)	89 (26.0)
Diabetes mellitus, n_0_ (%)	42 (13.2)
Chronic bronchitis, n_0_ (%)	219 (64.0)
COPD	149 (44.5)
GOLD COPD	
GOLD 1 (80% ≤ FEV1), n_0_ (%)	46 (31)
GOLD 2 (50% ≤ FEV1 < 80%), n_0_ (%)	90 (60.8)
GOLD 3 (30% ≤ FEV1 < 50%), n_0_ (%)	12 (8.1)
GOLD 4 (FEV1 < 30%), n_0_ (%)	0
FEV1 (% predicted), mean (SD)	83 (19.7)
FEV1/FVC (%), mean (SD)	70 (13.5)
FEV1 (% predicted)	
80% ≤ FEV1, n_0_ (%)	184 (53.9)
70% ≤ FEV1 < 80%, n_0_ (%)	81 (23.7)
60% ≤ FEV1 < 70%, n_0_ (%)	45 (13.2)
50% ≤ FEV1 < 60%	20 (5.8)
FEV1 < 50%, n_0_ (%)	11 (3.2)
Surgical procedures	
Lobectomy, n_0_ (%)	242 (70.1)
Bilobectomy, n_0_ (%)	20 (5.8)
Pneumonectomy, n_0_ (%)	83 (24.1)
Parietectomy, n_0_ (%)	19 (5.5)
Preoperative treatments	
Chemotherapy, n_0_ (%)	88 (22.9)
Radiotherapy, n_0_ (%)	14 (4.1)
Histological types	
Adenocarcinoma, n_0_ (%)	160 (47.9)
Squamous cell carcinoma, n_0_ (%)	133 (39.8)
Large-cell carcinoma, n_0_ (%)	25 (7.5)
Others +, n_0_ (%)	16 (4.8)
Pathological stage and tumor characteristics	
Pathological stages (4 classes)	
I, n_0_ (%)	138 (41.4)
II, n_0_ (%)	72 (21.6.)
III, n_0_ (%)	109 (32.7)
IV, n_0_ (%)	14 (4.2)
Pathological stages (8 classes)	
IA, n_0_ (%)	69 (20.7)
IB, n_0_ (%)	48 (14.4)
IIA, n_0_ (%)	45 (13.5)
IIB, n_0_ (%)	44 (13.2)
IIIA, n_0_ (%)	100 (30.0)
IIIB, n_0_ (%)	19 (5.7)
IIIC, n_0_ (%)	0
IV, n_0_ (%)	8 (2.4)
Morphomics and nutritional parameters	
Height, mean (SD) cm	170 cm (160–180)
Weight, mean (SD) kg	73.0 kg (57.8–88.2)
BMI	
Pre-surgery body mass index (BMI), mean (SD)	24.4 (4.4)
BMI < 18.5, n_0_ (%)	37 (10.4)
18.5 ≤ BMI < 25, n_0_ (%)	181 (51.2)
25 ≤ BMI, n_0_ (%)	135 (38.4)
% decrease in body weight at surgery, mean (SD)	4.0 (5.9)
Increased body weight, n_0_ (%)	11 (3.3%)
Stable or decreased body weight < 5%, n_0_ (%)	208 (62.5%)
5% ≤ Decreased body weight < 10%, n_0_ (%)	55 (16.5)
10% ≤ Decreased body weight, n_0_ (%)	59 (17.7)
Biological variable	
Nutritional variable	
Prealbumin, mean (SD) mg/mL	275 (92)
Prealbumin < 275 mg/mL, n_0_ (%)	203 (59.0)
275 mg/mL ≤ prealbumin, n_0_ (%)	141 (41.0)
Inflammatory variable	
CRP, mean (SD) mg/mL	22.8 (43)
CRP ≤ 3 mg/mL, n_0_ (%)	173 (50.3)
3 mg/mL< CRP, n_0_ (%)	171 (49.7)

SD: standard deviation. BC: chronic bronchitis. COPD: chronic obstructive pulmonary diseases. FEV1: forced expiratory volume 1 s. % predictive value. BMI: body mass index. CRP: C-reactive protein. + These include sarcomatoid carcinomas and adenosquamous carcinomas. The value in the table corresponds to the number of patients informed for a specific variable.

**Table 2 cancers-14-00874-t002:** Impact of clinical, pathological, and biological variables following univariate analysis on very long-term survival rates for the 345 study patients, obtained by Kaplan–Meier estimator, and comparison for each group of factors using the log-rank method censoring at 20 years.

	Univariate				
Variable	5-Year Survival Rate (95%CI)	10-Year Survival Rate (95% CI)	15-Year Survival Rate (95% CI)	20-Year Survival Rate (95% CI)	*p*
Global survival	41.9 (36.7–47.1)	23.8 (19.6–28.6)	12.2 (9.1–16.0)	5.7 (3.4–9.3)	
Sex					
Men	40.7 (35.2–46.4)	23.1 (18.6–28.3)	10.3 (7.3–14.4)	4.1 (2.0–7.9)	
Women	46.3 (33.7–59.4)	27.8 (17.6–40.8)	22.2 (13.2–34.9)	14.3 (6.7–27.6)	0.059
Patient age					
19–49	45.1 (32.3–58.6)	23.5 (14.0–36.8)	17.6 (9.6–30.2)	12.5 (5.7–25.3)	
50–59	46.6 (37.2–56.1)	29.1 (21.2–38.5)	17.5 (11.3–25.9)	9.7 (4.8–18.5)	
60–69	41.8 (33.4–50.7)	24.6 (17.8–32.9)	9.0 (5.1–15.4)	3.2 (1.1–9.1)	
70–84	35.1 (26.2–45.2)	16.0 (9.9–24.7)	3.2 (1.1–8.9)	1.1 (0.2–5.8)	0.0042
Smoking					
Never smoked	43.5 (25.6–63.2)	34.8 (18.8–55.1)	20.7 (6.9–37.1)	13.0 (4.5–32.1)	
Past or	42.1 (36.8–47.6)	23.0 (18.6–27.9)	10.1 (7.2–13.9)	5.3 (3.0–9.0)	0.24
Current smoker					
Smoking cessation before surgery					
Yes	42.0 (35.0–49.3)	23.2 (17.6–29.9)	12.2 (8.2–17.7)	6.9 (3.7–12.5)	
No	41.5 (33.5–49.9)	22.2 (16.0–29.9)	10.4 (6.3–16.7)	3.0 (0.9–8.9)	0.39
Cum. Smok.					
<20 p./yr	41.3 (34.4–48.5)	27.2 (21.3–34.0)	16.8 (12.1–22.9)	9.5 (5.7–15.4)	
20–50 p./yr	43.4 (38.2–48.8)	24.7 (20.3–29.6)	14.8 (11.3–19.0)	6.9 (3.7–12.7)	
50–100 p./yr	40.3 (33.5–47.5)	20.4 (15.3–26.8)	11.3 (7.5–16.6)	4.6 (1.9–10.3)	
100 p./yr≤	25.0 (8.9–53.3)	16.7 (4.7–44.8)	8.3 (1.5–35.9)	0	0.33
Com. Illness. Resp. status					
Alcohol					
No	44.4 (38.4–50.6)	26.6 (21.5–32.7)	14.3 (10.5–19.5)	7.5 (4.7–11.8)	
Yes	34.8 (25.7–45.7)	15.7 (9.6–24.7)	6.7 (3.1–13.9)	4.5 (1.7–10.9)	0.062
Diabetes mel.					
No	42.9 (37.2–48.8)	24.0 (19.3–29.9)	13.1 (9.6–17.6)	6.1 (3.5–10.1)	
Yes	31.0 (19.0–46.0)	19.0 (9.9–33.3)	4.8 (1.3–15.8)	2.4 (0.4–12.3)	0.31
Chron. bronch.					
No	46.3 (37.8–55.1)	26.8 (19.8–35.3)	17.1 (11.4–24.7)	9.9 (5.6–16.8)	
Yes	39.0 (32.7–45.6)	22.5 (17.4–28.4)	9.2 (6.0–13.8)	3.0 (1.1–8.3)	0.027
COPD					
No	43.9 (37.0–51.0)	22.8 (17.3–29.2)	13.2 (9.1–18.8)	7.9 (4.6–12.9)	
yes	38.8 (31.4–46.7)	24.3 (18.2–31.7)	11.2 (7.1–17.2)	2.5 (0.6–10.1)	0.40
COPD					
Mild	41.5 (29.2–54.9)	24.5 (14.9–37.5)	15.1 (7.8–27.0)	9.4 (4.1–20.2)	
Severe	42.1 (33.1–51.5)	29.9 (22.0–39.1)	15.9 (10.1–23.9)	4.0 (0.9–15.9)	0.97
FEV1 (% pred.)					
80%≤	46.9 (40.1–53.9)	27.6 (21.7–34.2)	17.9 (13.1–23.8)	13.6 (9.4–19.1)	
70–80%	40.4 (31.0–50.5)	28.7 (20.5–38.6)	14.9 (9.1–23.4)	7.0 (2.8–16.2)	
60–70%	39.3 (27.5–52.3)	30.4 (19.9–43.3)	14.3 (7.4–25.7)	10.7 (5.0–21.4)	
50–60%	34.8 (18.8–55.1)	14.5 (6.9–37.1)	13.0 (4.5–32.1)	8.7 (2.4–26.8)	
<50%	42.1 (23.1–63.7)	21.1 (8.5–43.3)	17.1 (5.5–37.0)	15.8 (5.5–37.5)	0.86
Preoperative treat.					
Chemotherapy					
No	47.4 (41.3–53.6)	26.3 (21.2–32.0)	13.5 (9.8–18.3)	5.9 (3.3–10.4)	
Yes	26.1 (18.0–36.1)	17.0 (10.6–26.2)	8.0 (3.9–15.5)	5.7 (2.4–12.6)	0.0076
Radiotherapy					
No	41.8 (36.6–47.2)	24.3 (19.9–29.2)	12.6 (9.4–16.6)	6.1 (3.6–9.8)	
Yes	28.6 (11.7–54.6)	7.1 (1.2–31.4)	0	0	0.071
Resection type					
Lobectomy	46.9 (40.7–53.1)	27.0 (21.7–32.9)	14.5 (10.6–19.5)	7.5 (4.4–12.3)	
Bilobectomy	40.0 (21.9–61.3)	30.0 (14.5–51.9)	10.0 (2.8–30.1)	5.0 (0.9–23.6)	
Pneumonec.	24.1 (16.1–34.3)	10.8 (5.8–19.3)	3.6 (1.2–10.1)	1.2 (0.2–6.5)	0.0000067
Pathological stage					
I	50.9 (44.4–57.4)	29.7 (24.1–36.0)	16.7 (12.3–22.1)	7.8 (4.4–13.2)	
II	26.1 (20.2–32.9)	14.4 (10.0–20.3)	6.1 (3.4–10.6)	5.0 (2.6–9.2)	
III	25.4 (18.5–33.8)	13.1 (8.2–20.2)	3.3 (1.3–8.1)	1.6 (0.4–5.8)	
IV	28.6 (11.7–54.6)	14.3(4.0–39.9)	7.1 (1.3–31.4)	0	0.00000042
Mediastinal pathological stage					
IA	68.1 (56.4–77.9)	40.6 (29.8–52.3)	23.2 (14.8–34.4)	6.5 (1.6–23.4)	
IB	64.6 (50.4–76.6)	33.3 (21.7–47.5)	16.7 (8.7–29.6)	4.2 (0.9–14.4)	
IIA	37.8 (25.1–52.4)	15.6 (7.7–28.8)	13.3 (6.2–26.2)	8.9 (3.0–23.3)	
IIB	29.5 (18.2–44.2)	20.5 (11.1–34.5)	6.8 (2.3–18.2)	6.8 (2.3–18.2)	
IIIA	25.3 (17.7–34.6)	12.1 (7.0–20.0)	3.0 (1.0–8.5)	2.0 (0.6–7.0)	
IIIB	21.1 (8.5–43.3)	10.5 (2.9–31.4)	0	0	
IV	12.5 (2.2–47.0)	0	0	0	<0.0000001
Histopath. type					
Adenocarci.	43.4 (35.9–51.2)	22.6 (16.8–29.7)	11.9 (7.8–17.9)	6.0 (3.0–11.4)	
Squam. cell carcinoma	42.1 (34.0–50.6)	26.3 (19.6–34.4)	10.5 (6.4–16.9)	1.9 (0.4–8.4)	
Large-cell carc.	26.0(17.2–47.5)	28.0(14.2–47.5)	8.0 (2.2–25.0)	8.0 (2.2–24.9)	
Others +	37.5 (18.5–61.4)	31.2 (14.2–55.6)	25.0 (10.2–49.5)	25.0 (10.2–49.5)	0.33
Morphomics					
Nutritionals par.					
Presurgical BMI					
BMI < 18.5	29.7 (17.4–45.7)	13.5 (5.9–27.9)	10.8 (4.2–24.7)	4.1 (0.8–17.9)	
18.5 ≤ BMI < 25	38.1 (31.3–45.3)	19.9 (14.7–26.3)	8.3 (5.0–13.2)	2.4 (0.7–7.1)	
25 ≤ BMI	49.6 (41.3–57.9)	30.4 (23.2–38.5)	16.3 (11.0–23.4)	11.1 (6.8–17.5)	0.0090
Body weight					
at surg. % var.					
increased	60 (31.2–83.3)	20 (5.6–50.9)	10 (1.8–40.4)	10 (1.7–40.4)	
Stable. or					
decreas. < 5%	46.9 (40.2–53.6)	29.0 (23.2–35.5)	14.5 (10.3–19.9)	8.1 (4.7–13.4)	
5 ≤ decreas. < 10	34.5 (23.3–47.7)	21.8 (12.9–34.3)	9.1 (3.9–19.6)	5.5 (1.9–14.8)	
10% ≤ decreas.	27.1 (17.4–39.6)	11.9 (5.9–22.5)	6.8 (2.6–16.2)	1.7 (0.3–9.0)	0.0034
Biological variables					
Nutrition varia.					
prealbumin					
preal. < 275 mg/mL	47.5 (39.4–55.7)	29.1 (22.2–37.0)	14.2 (9.3–20.9)	6.7 (3.4–12.7)	
275 ≤ prelab.	36.7 (30.4–43.5)	20.3 (15.4–26.2)	10.1 (6.7–15.0)	5.3 (2.7–10.1)	0.065
Inflammatory					
CRP ≤ 3 mg/mL	45.7 (38.3–53.4)	26.8 (20.6–34.1)	14.6 (10.0–20.8)	10.1 (6.3–15.8)	
3 mg/mL < CRP	37.8 (31.0–45.0)	21.1 (15.7–27.6)	10.0 (6.4–15.2)	2.4 (0.7–7.0)	0.049

+ These include sarcomatoid carcinomas, adenosquamous carcinomas, CNEGC. Pneumonec.: pneumonectomy. Histopathology type: Histopath. Type. Squam. Cell Carcinoma: squamous cell carcinoma. Large-cell carc.: large-cell carcinoma. Adenoc. Gr.: grade of adenocarcinoma. Adenocarcin.: adenocarcinoma. 95% CI: 95% confidence interval. *p*-value at 20 years obtained by Kaplan–Meier and log-rank method was used for comparing survival curves.

**Table 3 cancers-14-00874-t003:** Multivariate analysis of factors influencing risk of death at 20 years by Cox proportional hazards models. In this main model, model 1, the pathological stage in 7 classes, and model 2 correspond to model 1 adjust to preoperative FEV1%.

Variable	Relative Risk (RR)	95% CI of RR	*p*
*p* = 0.000075 model			
Number of patients at multivariate *n* = 276			
**Model 1**			
Pathological stages (7 classes)			
IA	1		
IB	1.12	(1.04–1.20)	
IIA	1.25	(1.08–1.44)	
IIB	1.39	(1.12–1.72)	
IIIA	1.55	(1.17–2.07))	
IIIB	1.73	(1.21–2.48)	
IV	1.93	(1.26–2.97)	0.0026
Age (years)			
<50 yrs	1		
50 yrs–60 yrs	1.21	(1.06–1.37)	
60 yrs–70 yrs	1.46	(1.13–1.89)	
70 yrs ≤	1.77	(1.20–2.60)	0.004
Cumulative smoking			
<20 p./yr	1		
20 p./yr–50 p./yr	1.21	(1.00–1.47)	
50 p./yr–100 p./yr	1.47	(1.01–2.15)	
100 p./yr≤	1.79	(1.01–3.16)	0.045
BMI (kg/m^2^)			
BMI < 18.5	1		
18.5 ≤ BMI < 25	0.82	(0.67–1.00)	
25 ≤ BMI	0.67	(0.45–0.99)	0.046
Chronic bronchitis			0.20
COPD yes/no			0.36
Sexe			0.28
Weight loss			
<5%			
5–10%			
10%≤			0.41
Diabete mellitus			
Yes/no			0.44
Type of resection			
Pneumonec./bilob. or lob.			0.60
Histological type			
Aden./Squa./Lar.cell/others			0.62
Alcohol abuse			
Yes/no			0.74
COPD GOLD			0.96
Past or current smoker			
Yes/no			0.99
**Variable**	**Relative Risk (RR)**	**95% CI of RR**	** *p* **
**Model 2**			
*p* = 0.00000059 model			
Number of patients at multivariate *n* = 276			
FEV1 preoperative (%)			
<50%	1		
50–60%	0.76	(0.66–0.86)	
60–70%	0.57	(0.44–0.74)	
70–80%	0.43	(0.29–0.63)	
80%≤	0.33	(0.19–0.54)	0.000019
COPD			
GOLD1	1		
GOLD2	0.67	(0.50–0.90)	
GOLD3	0.45	(0.25–0.81)	0.0079
Pathological stages (7 classes)			
IA	1		
IB	1.14	(1.06–1.22)	
IIA	1.29	(1.12–1.49)	
IIB	1.47	(1.18–1.83)	
IIIA	1.67	(1.24–2.23)	
IIIB	1.89	(1.31–2.73)	
IV	2.15	(1.39–3.34)	0.00063
Chronic bronchitis			
No	1		
Yes	1.40	(1.03–1.90)	
Age (years)			0.03
<50 yrs	1		
50 yrs–60 yrs	1.18	(1.04–1.35)	
60 yrs–70 yrs	1.40	(1.08–1.83)	
70 yrs<	1.65	(1.11–2.47)	0.013
BMI kg/m^2^			
BMI < 18.5	1		
18.5 ≤ BMI < 25	0.79	(0.65–0.96)	
25 ≤ BMI	0.62	(0.42–0.93)	0.02
Cumulative smoking			
<20 p./yr	1		
20 p./yr–50 p./yr	1.18	(0.98–1.43)	
50 p./yr–100 p./yr	1.39	(0.95–2.04)	
100 p./yr≤	1.65	(0.93–2.91)	0.087
Weight loss			
<5%			
5–10%			
10%≤			0.20
Histological type			
Aden./Squa./Lar.cell/others			0.32
Sexe			0.39
Diabete mellitus			
Yes/not			0.44
Type of resection			
Pneumonec./bilob. or lob.			0.65
COPD yes/no			0.73
Past or current smoker			
Yes/no			0.87
Alcohol abuse			
Yes/no			0.99

The relative risks are reported only for variable with a *p*-value ≤ 0.1. These include sarcomatoid carcinomas and adenosquamous carcinomas. IIIC patients with N3 diseases are not ever operated. GOLD4 patients with preoperative VEMS below 30% are not operable.

## Data Availability

The data presented in this study are available in this article (and Appendix A).

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
