# Peer review of "Twenty-Year Survival of Patients Operated on for Non-Small-Cell Lung Cancer: The Impact of Tumor Stage and Patient-Related Parameters"

_cancers, 2022, doi:10.3390/cancers14040874_

Round 1
Reviewer 1 Report
The Authors reported the results of retrospective analysis conducted on a prospectively collected database of 345 patients who underwent a major lung resection for a lobectomy or pneumonectomy between June 15, 2001 and January 2002 for NSCLC in a single high-volume University Hospital. They found that many clinical factors that predict 5-/10-year survival, such as pathological stage, chronic bronchitis, chronic obstructive diseases (COPD), and pulmonary function as well as nutritional parameters are also determining 20-year outcomes. The results are interesting and well presented. I just suggest to delete the subheadings in the discussion.
Author Response
Reviewer: The Authors reported the results of retrospective analysis conducted on a prospectively collected database of 345 patients who underwent a major lung resection for a lobectomy or pneumonectomy between June 15, 2001 and January 2002 for NSCLC in a single high-volume University Hospital. They found that many clinical factors that predict 5-/10-year survival, such as pathological stage, chronic bronchitis, chronic obstructive diseases (COPD), and pulmonary function as well as nutritional parameters are also determining 20-year outcomes. The results are interesting and well presented.
Reply: We would like to thank the Reviewer for kind appreciation of our work.
Reviewer:I just suggest to delete the subheadings in the discussion.
Reply: As suggested we removed subheadings in the discussion.
Reviewer 2 Report
I appreciate your work, it's originality and the quality of the statistical analysis. I think that you used the old TNM classification for lung cancer and surgical staging because of the very long period of analysis for the standardization of the results.

Author Response
Reviewer: I appreciate your work, it's originality and the quality of the statistical analysis.
Reply: We would like to thank the Reviewer for kind appreciation of our work.
Reviewer: I think that you used the old TNM classification for lung cancer and surgical staging because of the very long period of analysis for the standardization of the results.
Reply. When pathological review was made, TNM staging was reattibuted by the two pathologists. Effectively stages were reattributed according to the 7th system staging. This is better stated in the revised version.
Reviewer 3 Report
This study was concerned with the evaluation of impact of surgery on 15- and 20-year survival in patients with non-small cell lung cancer. A total of 293 patients operated for major-lung-resection were enrolled in this study. It was concluded that the very-long-term survivals could be achieved after surgery of NSCLC and factors classically predicting 5- and 10-years survival also determined longer outcomes.
As authors mentioned that the main limitation of this study was being a small-scale single center study. Because of those limitations, there was not any exciting finding in this study. For example, the risk factors predicting 15- and 20-years survival were found similar to those for 5- and 10-year survival and the rate of declining in survival in patients with NSCLC were similar to people without tumors. Overall, it seems that the 15- and 20-years survival of patients with NSCLC is of limited clinical relevance.
Author Response
Reviewer: This study was concerned with the evaluation of impact of surgery on 15- and 20-year survival in patients with non-small cell lung cancer. A total of 293 patients operated for major-lung-resection were enrolled in this study. It was concluded that the very-long-term survivals could be achieved after surgery of NSCLC and factors classically predicting 5- and 10-years survival also determined longer outcomes.
As authors mentioned that the main limitation of this study was being a small-scale single center study. Because of those limitations, there was not any exciting finding in this study. For example, the risk factors predicting 15- and 20-years survival were found similar to those for 5- and 10-year survival and the rate of declining in survival in patients with NSCLC were similar to people without tumors. Overall, it seems that the 15- and 20-years survival of patients with NSCLC is of limited clinical relevance.
Reply: we thank the Reviewer for evaluation of our study. As underlined by the reviewer, our study has some limitations underlined in the specific paragraph of the discussion. However, as survival beyond 15 years has never been reported so far, we do think that our study will provide a significant contribution in knowledges about lung cancer treatments.